# Genetic Interaction of Global Regulators AflatfA and AflatfB Mediating Development, Stress Response and Aflatoxins B1 Production in *Aspergillus flavus*

**DOI:** 10.3390/toxins14120857

**Published:** 2022-12-04

**Authors:** Xiuna Wang, Wenjie Zha, Bin Yao, Lan Yang, Shihua Wang

**Affiliations:** Key Laboratory of Pathogenic Fungi and Mycotoxins of Fujian Province, College of Life Sciences, Fujian Agriculture and Forestry University, Fuzhou 350002, China

**Keywords:** *Aspergillus flavus*, bZIP transcription factor, AflatfA, AflatfB, aflatoxin B1, development, stress response

## Abstract

*Aspergillus flavus* produces carcinogenic and mutagenic aflatoxins, which cause economic losses and risk of food safety by contaminating grains, food and feed. In this study, we characterized two bZIP transcription factors, AflatfA and AflatfB, and their genetic interaction. Compared to the wild type (WT), *AflatfA* deletion and *AflatfA* and *AflatfB* double deletion both caused retarded vegetative growth of mycelia. Relative to WT, the *AflatfA* deletion strain (Δ*AflatfA*) and *AflatfA* and *AflatfB* double deletion strain (Δ*AflatfA*Δ*AflatfB*) produced more sclerotia, whereas the *AflatfB* deletion strain (Δ*AflatfB*) produced less sclerotia. After 4 °C preservation and incubation at 50 °C, conidia viability dramatically decreased in the Δ*AflatfA* and Δ*AflatfA*Δ*AflatfB* but Δ*AflatfB* mutants, whereas conidia viability of the Δ*AflatfA*Δ*AflatfB* strain was higher after storage at 4 °C than in *AflatfA* mutant. Conidia of Δ*AflatfA*, Δ*AflatfB* and Δ*AflatfA*Δ*AflatfB* strains significantly increased in sensitivity to H_2_O_2_ in comparison with WT. Compared to WT, the mycelium of Δ*AflatfA* and Δ*AflatfB* strains were more sensitive to H_2_O_2_; conversely, the Δ*AflatfA*Δ*AflatfB* strain showed less sensitivity to H_2_O_2_. Δ*AflatfA* and Δ*AflatfA*Δ*AflatfB* strains displayed less sensitivity to the osmotic reagents NaCl, KCl and Sorbitol, in comparison with WT and Δ*AflatfB* strains. When on YES medium and hosts corn and peanut, Δ*AflatfA* and Δ*AflatfA*Δ*AflatfB* strains produced less aflatoxin B1 (AFB1) than Δ*AflatfB*, and the AFB1 yield of Δ*AflatfB* was higher than that of WT. When WT and mutants were inoculated on corn and peanut, the Δ*AflatfA* and Δ*AflatfA*Δ*AflatfB* but not Δ*AflatfB* mutants produced less conidia than did WT. Taken together, this study reveals that AflatfA controls more cellular processes, and the function of AflatfA is stronger than that of AflatfB when of the same process is regulated, except the response to H_2_O_2_, which might result from the effect of AflatfA on the transcriptional level of AflatfB.

## 1. Introduction

Several transcription factors (TF) as global regulators are found to regulate secondary metabolism, virulence, stress response and development [1]. Among them, basic leucine zipper (bZIP) transcription factors are critical global regulators [2]. The family of the bZIP transcription factor is not only one of the largest but also the most conserved TF families across eukaryotes [3]. This family of transcription factors contains the conserved basic leucine zipper domain, which is responsible for dimerization, DNA binding and nuclear import. bZIP transcription factors are classified into various subfamilies, such as the ATF/CREB (activating transcription factor/cAMP-responsive element-binding protein) family. Among them, Atf1/AtfA is one of most well-studied TFs. As is known, Atf1/AtfA acts downstream of the stress-activated mitogen-activated protein kinase (SAPK) cascade. The function and regulatory mechanism of Atf1 protein from fission yeast has been studied extensively. Activated and stabilized via phosphorylation by Sty1, the transcription factor Atf1 regulates genes involved in stress responses [4]. In *Schizosaccharomyces pombe*, Atf1 lowers the transition barrier for nucleosome-mediated establishment of heterochromatin, activates the *fbp1* gene transcription induced upon glucose starvation and modulates meiotic recombination and heterochromatin formation [5,6,7,8,9,10]. Atf1 frequently forms heterodimers with other bZIP TF Pcr1, and the interaction of Atf1 and Pcr1 regulates not only stress response genes but also cyclin expression during G2/M transition [8,11,12]. Moreover, Atf1 is a target for the ubiquitin-proteasome system and that its degradation is dependent upon an SCF E3 ligase containing the F box protein Fbh1 [13]. The binding of the transcription factor Atf1 to promoters serves as a barrier to phase nucleosome arrays and avoids cryptic transcription [14].

Orthologs of *S*. *pombe* Atf1are conserved in filamentous fungi, however, the functions and regulatory mechanisms are heterogeneous, including regulation of vegetative growth and development, a broad spectrum of stress responses, secondary metabolism and virulence. Deletion of *atf1* affects vegetative growth in *Aspergillus nidulans*, *Aspergillus oryzae*, *Botrytis cinerea*, *Magnaporthe oryzae* and *Fusarium verticillioides* [15,16,17,18,19]. Atf1/AtfA is found to regulate the sexual development of several fungi, including *Neurospora crassa* and *Fusarium graminearum* [20,21,22]. Many Atf1/AtfA orthologs, from *A*. *nidulans*, *Aspergillus fumigatus*, *F*. *verticillioides* and *Verticillium dahliae*, are involved in conidium production [15,19,23]. In *Claviceps purpurea* and *B*. *cinerea*, the formation of sclerotia is also mediated by Atf1/AtfA orthologs [17,24]. AtfA transcription factors also contribute to a variety of stress responses. The response of AtfA to oxidative stress is most universal in fungi, including *A*. *nidulans*, *A*. *fumigatus*, *A*. *oryzae*, *Penicillium marneffei* (*Talaromyces marneffei*), *M*. *oryzae*, *F*. *graminearum*, and *F*. *verticillioides* and *Fusarium oxysporum* [16,18,19,21,22,25,26,27,28,29,30,31,32]. The coordination of AtfA orthologs with oxidative stress response of fungi is related with both types of chemical reagents and morphological forms of fungi. An osmotic stress-sensitive phenotype has also been characterized in fungi, including *N*. *crassa*, *A*. *nidulans*, *A*. *fumigatus* and *F*. *graminearum* [20,21,22,27,33]. It is noteworthy that Atf1/AtfA orthologs, from *N*. *crassa*, *A*. *nidulans* and *A*. *fumigatus*, are also involved in the fungicide resistance [33,34,35]. Moreover, Atf1/AtfA orthologs are critical in cell wall integrity in *A*. *fumigatus*, *F*. *graminearum*,and *F*. *verticillioides* [19,22,36]. Furthermore, conidia of Δ*AtfA* strains show decreased viability after long-term storage at 4 °C as well as heat treatment in *A*. *nidulans* and *A*. *fumigatus* [26,27]. In addition, AtfA is involved in the response of conidia to heavy metal in *A*. *nidulans* [37], and AtfA affects conidia germination after dessication under vacuum in *A*. *fumigatus* [27]. A novel function is revealed that nitrosative stress response is controlled by AtfA in *V*. *dahliae* [38]. Atf1/AtfA orthologs, as global regulators, are also important players in the regulation of secondary metabolite biosynthesis in *A*. *nidulans*, *A*. *fumigatus* and *Aspergillus terreus*, *Mucor circinelloides*, *B*. *cinerea*, *F*. *graminearum* and *F*. *verticillioides* [17,19,21,22,29,39,40,41]. However, the regulation mechanism is complex, which is not only related with fungal species but also types of secondary metabolites and the host of pathogens. Atf1/AtfA orthologs as critical virulence factord affect pathogenicity in almost all pathogenic fungi, including human pathogens (*A*. *fumigatus*, *P*. *marneffei*, *M*. *circinelloidesa*, and *Cryptococcus neoformans*) and plant pathogens (*B*. *cinerea*, *F*. *graminearum*, *F*. *verticillioides*, *M*. *oryzae*, *C*. *purpurea* and *F*. *oxysporum*) [15,17,18,19,21,22,24,27,31,32,36,41,42]. 

Among the bZIP/CREB family, a novel homologous gene is firstly found in *A*. *oryzae*, which is shorter in the N-terminal region than AtfA [43]. *A*. *oryzae atfB* regulates the response of conidia and mycelium to H_2_O_2_ stress and conidia formation under hyperosmotic stress, whereas the conidia of Δ*atfB* increase in sensitivity to heat-shock [43]. Roze et al. (2004) found that a novel cAMP-response element, CRE1, modulates expression of the aflatoxin biosynthetic gene nor-1 in *Aspergillus parasiticus*, and that protein p32 with an approximate molecular mass of 32 kD binds CRE1 and physically interacts with the aflatoxin pathway regulator AflR [44]. A further study reveals that this protein shares 96% of its identity with the *A*. *oryzae* bZIP protein AtfB, and *atfB* is not only related with stress response but also integrates secondary metabolisms with an oxidative stress response in *Aspergilli* [45]. AtfB binds at the aflatoxin biosynthetic gene promoters that carry a CRE motif, and the five nucleotides upstream from CRE1, AGCC(G/C) are highly conserved [45]. The expression of the gene *atfB* is positively influenced by a master regulator *veA*, and the normal formation of DNA-protein complexes in the *cat1* promoter is dependent on AtfB in *A*. *flavus* [46]. By gene silencing, AtfB controls virulence-associated processes in *A*. *parasiticus* [47]. 

Mycotoxins cause serious harm to human health by food or feed safety [48,49,50]. Among them, aflatoxins are the most notorious, especially AFB1, which is one of the most carcinogenic and mutagenic natural producers [51]. It is known that the opportunistic phytopathogen *Aspergillus flavus* is the main producer of aflatoxins, which infects oil seed crops such as corn, peanut, tree nuts and cotton [52]. Importantly, *A*. *flavus* is also an opportunistic human pathogen [52]. *A*. *flavus’s* pathogenicity and aflatoxins-producing capability are coupled to environmental stress response, spore production, vegetative growth and so on [53]. Therefore, it is an advanced strategy to control *A*. *flavus* and aflatoxins by environment and development [51]. In *A*. *flavus*, RNA-seq analysis indicates that AtfA could regulate the inhibition of piperine on aflatoxins B1 production by modulating fungal oxidative stress response [54], and expression of AtfA is significantly decreased by the inhibitor of aflatoxins B1 epigallocatechin gallate (EGCG) [55]. Meanwhile, transcriptional level analyses demonstrate that AtfB could regulate different water activities and both processes of the inhibition of methyl jasmonate and inhibition of piperine on aflatoxins B1 in *A*. *flavus* [54,56,57].

In the *A*. *flavus* strain NRRL3357, AtfA and AtfB have impacts on growth, conidiation production, sclerotia formation, aflatoxin biosynthesis on artificial medium and oxidative stress response, whereas only AtfA affects cell wall stress response and pathogenicity [58]. Under stress, the *atfA* gene is controlled by the HogA (SakA) SAPK pathway in *A*. *nidulans* and *A*. *fumigatus* [33,35]. Intriguingly, deletion of *AtfA* has no effect on osmotic stress response in the *A*. *flavus* strain NRRL3357 [58]. Meanwhile, the genetic interaction of AflatfA and AflatfB is not clear. In this study, *AflatfA* and *AflatfB* were firstly deleted in the recipient strain (PTSΔ*ku70*Δ*pyrG*), and then *AflatfA* and *AflatfB* were complemented and *AflatfB* was deleted in the *AflatfA* strain to construct the double deletion strain. To characterize the functions of AtfA and AtfB in *A*. *flavus*, unlike the deletion strain containing the RFP gene and recipient strain NRRL3357 without the RFP gene [58], we compared the deletion strains with the WT strain but not the recipient strain, both of which contain the selective gene *pyrG*. We revealed the genetic interaction of AflatfA and AflatfB, and found several novel and different functions. Our results indicate that AflatfA positively regulates the transcription of the *AflatfB* gene, and more biological processes are meditated by *AflatfA* than by *AflatfB*, whereas *AflatfA* plays a decisive role and *AflatfA* and *AflatfB* have no overlapping functions.

## 2. Results

### 2.1. AflatfA and AflatfB Are ATF/CREB Homologous Genes in A. flavus

Using the ATF/CREB homologs AtfA and AtfB amino acid sequences of *A*. *oryzae* as the query, respectively, the searches of the *A*. *flavus* genome database were performed, and the genes AFLA_031340 and AFLA_094010 were identified with the greatest similarity to AtfA (66.87%) and AtfB (99.69%), respectively. Then, AFLA_031340 and AFLA_094010 in *A*. *flavus* were named AflatfA and AflatfB, respectively. Bioinformatic analysis showed that AFLA_031340 encodes 512 amino acids with two introns (438 bp and 53 bp), and AFLA_094010 encodes 318 amino acids with no intron. The re-sequence of *AflatfA* ORF and cDNA identified that the *AflrsmA* gene consists of two introns, 53 and 438 bp. Moreover, a highly conserved bZIP domain was in the C-terminal of the Atf1 homologs, and OSM, HRA and HRR domains were conserved in AflatfA in comparison with the Atf1 homologs in filamentous fungi except *B*. *cinerea* and budding yeast *S*. *pombe* (Figure 1). There is only one bZIP domain in the AtfB from *A*. *flavus* and the compared fungi, including filamentous fungi and budding yeast. The phylogenetic tree based on the bZIP domain sequence showed that AflatfA is the most closely related to the AtfA homologs bZIP from *A*. *oryzae* (Figure 1).

### 2.2. The Generation of Mutants and Expression of AflatfB Was Impacted by AflatfA

In order to study the possible functions of AflatfA and AflatfB in *A*. *flavus* using reverse forward genetics, *AflatfA* and *AflatfB* were firstly deleted. Then, using a gene-replacement approach, the *AflatfA* and *AflatfB* genes were deleted via PEG-mediated transformation of the *A*. *flavus* strain protoplast (Figure 2A). The putative knockout strains were then screened by diagnostic polymerase chain reaction (PCR). Furthermore, the putative deleted mutants were confirmed by Southern blot. When hybridized with a probe amplified from the homologous arm of *AflatfA* or *AflatfB* r, two *AflatfB* mutants showed an expected signal, but among the two *AflatfA* mutants, one T1 showed an expected signal, whereas the other had two more brands besides the expected 8533 bp brand (Figure 2B,C). The result indicated that there is an ectopic integration in the *AflatfA* mutant T2. Furthermore, the complementary strains were created by reintroducing the *AflatfA* or *AflatfB* ORF sequence into *AflatfA* T1 and *AflatfB* T1 (Figure 2D,E). To investigate the relationship of *AflatfA* and *AflatfB*, the *AflatfB* gene was replaced by the selected gene *pTRA*, through transformation of the *AflatfA* deletion strain T1 (Figure 2A). Diagnostic PCR confirmed that *pTRA* was integrated in the *AflatfB* locus in the genome of the *AflatfA* mutant (Figure 2F,G). 

To provide the insight of the relation between *AflatfA* and *AflatfB*, the transcription level was determined by RT-PCR. Compared with the WT strain, the transcriptional level of AflatfB was lower in the genome of the *AflatfA* deletion strain, but deletion of *AflatfB* has no impact on the transcriptional level of *AflatfA* (Figure 2H). Thus, *AflatfA* positively regulates the expression of *AflatfB* in *A*. *flavus*.

### 2.3. The Impact of AflatfA and AflatfB on Vegetative Growth and Sclerotia Formation of A. flavus

To investigate the function of AflatfA and AflatfB on mycelium growth in *A*. *flavus*, WT, Δ*AflatfA*, Δ*AflatfB*, Δ*AflatfA*Δ*AflatfB* and complementary strains were simultaneously incubated on the PDA medium. When grown on the PDA medium after five days, Δ*AflatfA* and Δ*AflatfA*Δ*AflatfB* were both significantly reduced in mycelium growth (Figure 3A,B) compared with the WT, Δ*AflatfB* and complementary strains. As a result, AflatfA positively regulates vegetative growth, whereas AflatfB is not involved in the regulation of colony growth. 

To survive in adverse environmental conditions, *A*. *flavus* produces the resistant structure sclerotia. Therefore, we investigated whether sclerotia formation was affected by *AflatfA* and *AflatfB*. WT, *AflatfA* and *AflatfB* mutants were inoculated on a modified Wickerham medium, and then the number of sclerotia was counted after 12 d. Compared to WT and corresponding complementary strains, the *AflatfA* deletion strain produced more sclerotia; however, the number of sclerotia from the Δ*AflatfB* strain had significantly decreased (Figure 3C,D). Interestingly, the number of sclerotia produced by the Δ*AflatfA*Δ*AflatfB* strain was significantly increased, which is similar to that in the *AflatfA* mutant (Figure 3C,D). The above results indicate that AflatfA negatively regulates sclerotia formation, but AflatfB positively regulates sclerotia formation, and AflatfA has a stronger function than AflatfB for sclerotia formation.

### 2.4. The Decline of AFB1 Yield from ΔAflatfA, ΔAflatfB and ΔAflatfAΔAflatfB Strains on Artificial Medium YES

To confirm whether *AflatfA* and *AflatfB* regulate the biosynthesis of aflatoxins, equal amounts of spore from WT and mutant strains were inoculated on a YES medium and cultured at 29 °C for 5 d in the dark. TLC analysis showed that the yield of aflatoxin AFB1 produced by Δ*AflatfA*, Δ*AflatfB* and Δ*AflatfA*Δ*AflatfB* strains were significantly lower than that by WT, but that of *AflatfB* mutant was significantly higher than that of Δ*AflatfA* and Δ*AflatfA*Δ*AflatfB* strains (Figure 4). Interestingly, there is no difference in AFB1 production between Δ*AflatfA* and Δ*AflatfA*Δ*AflatfB* strains (Figure 4). The data showed that AflatfA and AflatfB both positively regulate the biosynthesis of AFB1, and the function of AflatfA is more powerful than that of AflatfB. However, *AflatfA* and *AflatfB* have no additive effect on AFB1 production.

### 2.5. The ΔAflatfA Strain Has a Defect in the Conidia Response to Temperature Stress

The bZIP TF AtfA homologous genes from the *A*. *nidulans* and *A*. *fumigatus* function in the conidia response to heat stress and 4 °C preservation [26,27], and the contribution of *AflatfA* and *AflatfB* to the conidia response to temperature stress were determined. The conidia of WT and mutants were preserved at 4 °C for 7 d and 14 d, and then conidia viability was tested by the germinating rate. The viability of Δ*AflatfA* and Δ*AflatfA*Δ*AflatfB* strains were both significantly lower than that of the WT strain (Figure 5A,B). However, the conidia viability at 4 °C preservation showed no change among Δ*AflatfB*, WT and the two complementary strains (Figure 5A,B). Thus, AflatfA but not AflatfB mediates the conidial viability after long-term storage at 4 °C.

To investigate the functions of *AflatfA* and *AflatfB* in response to heat stress, the conidia of WT and mutants were treated at 50 °C for 1 h, and then the germinating rate was analyzed. The optimum growth temperature at 29 °C and heat stress both could result in the germinating rate of Δ*AflatfA* and Δ*AflatfA*Δ*AflatfB* conidia significantly being reduced compared with that of the WT strain (Figure 5C,D). However, heat stress made the germinating rate of WT conidia drop by 4.5%, whereas that of Δ*AflatfA* and Δ*AflatfA*Δ*AflatfB* conidia were reduced by up to 79.2% and 77.7%, respectively (Figure 5C,D). Compared to that of WT, the germinating rate of the conidia of Δ*AflatfB* and the two complementary strains were not different (Figure 5C,D). These observations implied that AflatfA but not AflatfB is required for response to heat stress in *A*. *flavus*. 

### 2.6. Deletion of AflatfA and AflatfB Increases Sensitivity to Oxidative Stress

The bZIP transcription factors AtfA and AtfB play important roles in response to oxidative stress in *Aspergillus* spp. [26,30,33]. The cause of the H_2_O_2_ inhibitory action on *A*. *flavus* may be that H_2_O_2_ impact conidia germination. To test this hypothesis, conidia were firstly treated by 200 mM H_2_O_2_ for 20 min, and then two hundred spores were inoculated on YES to examine the germinating rate of conidia. Without H_2_O_2_, the germinating rate of Δ*AflatfA* and Δ*AflatfA*Δ*AflatfB* strains significantly decreased compared to WT; however, the germinating rate of Δ*AflatfB* and WT had no significant difference (Figure 6A,B). After H_2_O_2_ treatment, the germinating rate of Δ*AflatfA* and Δ*AflatfA*Δ*AflatfB* reduced by 42% and 44% respectively, whereas the germinating rate of WT reduced by 32% (Figure 6A,B). Germinating rate assays demonstrated that Δ*AflatfB* conidia was more sensitive to H_2_O_2_ than WT (Figure 6A,B). Therefore, *A*. *flavus* AflatfA and AflatfB are both required for oxidative stress tolerance in conidia.

To study mycelia tolerance to oxidative stress, two hundred spores per strain were inoculated on the 2 mm cellophane plated on the YES medium. After 24 h, the cellophane containing spores were transferred to the YES medium containing 10 mM H_2_O_2_. As determined by the inhibition rate, Δ*AflatfA* and Δ*AflatfB* strains were more sensitive to H_2_O_2_ than to the WT strain (Figure 6C,D). Intriguingly, the sensitivity of the Δ*AflatfA*Δ*AflatfB* strain was lower than that of WT. However, there were no differences in inhibition rates among WT, Δ*AflatfA^C^* and Δ*AflatfB^C^* strains (Figure 6C,D). Taken together, AflatfA and AflatfB play critical roles in mycelium respose to oxidative stress. 

### 2.7. AflatfA but AflatfB Contributes to Osmotic Stress Response

As is known, bZIP transcription factor AtfA is a target gene that is regulated by the HOG pathway and plays an important role in the response to osmotic stress. To study the function of *AtfA* and *AtfB* homologs in *A*. *flavus* response to osmotic stress, WT and mutant strains were cultured on a GMM medium containing various osmotic reagents. Without an osmotic reagent, the colony diameter of Δ*AflatfA* and Δ*AflatfA*Δ*AflatfB* strains were smaller than that of the WT strain (Figure 7). Compared to that of WT, the colony diameter of Δ*AflatfA* and Δ*AflatfA*Δ*AflatfB* were significantly increased with 1.5 M NaCl, 1.8 M KCl or 1.4 M Sorbitol (Figure 7). However, there is no difference in colony diameter between Δ*AflatfB* and WT strains with or without osmotic reagents (Figure 7). The above results showed that *AflatfA* but not *AflatfB* regulates the osmotic stress response in *A*. *flavus*.

### 2.8. Effect of AflatfA and AflatfB on Pathogenicity of A. flavus

We performed topical infection with corn and peanut to examine the effect of *AflatfA* and *AflatfB* on *A*. *flavus* pathogenicity. After infecting corn and peanut for 5 d, the sporulation and AFB1 yield were estimated and compared among the WT and mutant strains. Compared to the WT strain, Δ*AflatfA* and Δ*AflatfA*Δ*AflatfB* produced less conidia (Figure 8A,B). However, the difference in sporulation was not significant between WT and Δ*aflatfB* (Figure 8A,B). These results indicated that *AflatfA* but not *AflatfB* positively regulates conidia production.

The pattern of AFB1 production was not the same as that of sporulation. Relative to the WT and complementary strains, the yield of aflatoxin AFB1, produced by Δ*AflatfA*, Δ*AflatfB* and Δ*AflatfA*Δ*AflatfB* strains, significantly decreased (Figure 8C,D). When comparing AFB1 produced by the deletion strains, we found that the Δ*AflatfB* strain produced the most mycotoxin AFB1, followed by Δ*AflatfA* and Δ*AflatfA*Δ*AflatfB* (Figure 8C). Statistical analysis of AFB1 production showed that there was significant difference between Δ*AflatfA* and Δ*AflatfB* but no significant difference between Δ*AflatfA* and Δ*AflatfA*Δ*AflatfB* (Figure 8D). Thus, *AflatfA* and *AflatfB* are both important for AFB1 production, but *AflatfA* plays a more important role than *AflatfB*.

## 3. Discussion

Our interest in exploring the function of the CRE/ATF family AflatfA (ortholog Atf1/AtfA) and AflatfB on AFs biosynthesis in *A*. *flavus* arose from the known functions of Atf1/AtfA in the plant pathogenic fungus *F*. *graminearum* and AtfB in aflatoxin producer *A*. *parasiticus* [21,22,45]. In *F*. *graminearum*, *Fgatf1* deletion leads to the production of higher amounts of mycotoxins DON and aurofusarin culturing in vitro, whereas Δ*Fgatf1* produced lower amounts of DON during the infection of the head of host wheat [21]. However, using another Δ*Fgatf1* mutant, Jiang et al. (2015) found that DON production was significantly reduced in the wheat head but no change was observed when using autoclaved rice grains cultures [22]. Conversely, in *F*. *verticillioides*, the regulation of *FvatfA* on the secondary metabolism is complex. *FvatfA* positively regulates the production of antioxidants carotenoids and mycotoxins fumonisin B1 and B2, whereas *FvatfA* deletion results in the yield of the red pigment bikaverin with anticancer and antimicrobial activities and increases without a change to *bik1* expression [19]. Temme et al. (2012) found that *Bcatf1* negatively regulates biosynthesis of secondary metabolites, including botrydial, botryendial and botcinin [17]. Similar to the known AtfA/Atf1 in other plant pathogenic fungi, *AflatfA* in *A*. *flavus* mediates AFB1 production. In contrast to the regulation of DON by *Fgatf1* in *F*. *graminearum* related with the host type and artificial medium [21,22], regulation of AflatfA on AFB1 production is independent on culture substrates, and the Δ*AflatfA* strain produced decreased AFB1 not only on the YES medium but also on host plants such as peanut and maize. Similar to the lesser amounts of the antioxidants carotenoids and mycotoxins fumonisin B1 and B2 produced by the *FvatfA* deletion strain [19], *AflatfA* deletion led to a decline in the AFB1 yield from the artificial medium and the host peanut and maize in this study. Zhao et al. (2022) found Δ*AtfA* produces reduced AFB1 on an artificial medium in *A*. *flavus* NRRL3357 [58]. Based on the above result, we speculate that AflatfA might bring about AFB1 production change by positively regulating AFB1 biosynthesis genes. The various functions of AftA/Atf1 orthologs from different fungi suggest that the role of AtfA/Atf1 in regulating fungal secondary metabolism is conserved but the regulation mechanism is divergent. However, there are fewer reports on the effect of AtfB, another member of the CRE/ATF family on secondary metabolism. In *A*. *flavus*, AtfB could down-regulate AFs biosynthesis at the transcription level analysis under inhibition condition of AFs biosynthesis [54,56,57]. Consistent with transcription level analysis, deletion of AtfB encoded by *A*. *flavus* genome results in a decline of AFB1 from an artificial medium [58]. In this study, we found the *AflatfB* deletion results in a decline in AFB1 yield in both the artificial medium and the host peanut and maize. Interestingly, the AtfB ortholog positively regulates AFB1 biosynthetic genes by binding the promoters that carry a CRE motif, and the five nucleotides immediately upstream from CRE1, AGCC(G/C) in *A*. *parasiticus* [44]. The AtfB protein sequence from *A*. *parasiticus* shares 96% of its identity with the amino acid sequence of *A*. *flavus* AtfB, suggesting that AtfB may directly bind the promoter of AFs biosynthesis [45]. In this study, AftA/Atf1 and AtfB orthologs from *A*. *flavus* both regulate AFB1 biosynthesis; however, AtfA develops a greater effect, when comparing the AFB1 yield of Δ*AflatfA*, Δ*AflatfB* and Δ*AflatfA*Δ*AflatfB* strains. 

In various of fungi, AtfA/Atf1 and AtfB from the ATF/CREB family of bZIP transcription factors were found to couple biosynthesis of secondary metabolites with the stress response of fungi, such as *AtfA* in *A*. *nidulans* and AtfB in *A*. *parasiticus* [25,28,29,45]. The conidia of Δ*AflatfA*, Δ*AflatfB* and Δ*AflatfA*Δ*AflatfB* were more sensitive to H_2_O_2_, which is similar to AtfA in *S*. *pombe*, *A*. *oryzae*, *A*. *nidulans*, *A*. *fumigatus* and *M*. *oryzae* [11,18,26,27,43]. The abovementioned data indicated that *AtfA* and *AtfB* orthologs may have a conserved role in regulating the response of conidia to oxidative stress. Analysis of mycelium sensitivity to oxidative stress reveals that Δ*AflatfA* and Δ*AflatfB* strains are more sensitive to H_2_O_2_ than WT, and the sensitivity of Δ*AflatfB* strains to H_2_O_2_ is higher than Δ*AflatfA*. Therefore, AtfA and AtfB both positively regulate the oxidative stress response in fungal mycelium. Intriguingly, double deletion of *AflatfA* and *AflatfB* results in more tolerance to H_2_O_2_ in comparison with WT. We assume that the other regulators are activated after double deletion of *AflatfA* and *AflatfB*. When conidia were inoculated on a medium containing the oxidative agent H_2_O_2_, Δ*AflatfA* and Δ*AflatfB* strains are both more sensitive to oxidative stress than WT [58], as was reported in *M*. oryzae, *F*. *graminearum* and *F*. *verticillioide* [18,19,21,22], but contrary to what was reported in *A*. *oryzae* and *A*. *nidulans* [16,25,28,29]. In addition, Atf1/AtfA homologs in *P*. *marneffei*, *V*. *dahliae* and *B*. *cinerea* have no effect on oxidative stress (H_2_O_2_) [17,23,31], but *atfA* in *P*. *marneffei* participates in oxidative stress (tBOOH) [31]. We deduce that the function of *atfA* in response to oxidative stress is related with not only species but also agent type. Thus, AflatfA and AflatfB have a positive oxidative stress response.

In the model filamentous fungus *A*. *nidulans*, the AtfA transcription factor interacts with the Sak/HogA MAPK and regulates various stress responses [28]. As the target gene of the Sak/HogA pathway, Atf1/AtfA orthoglogs mediate osmotic stress response in several fungi, including *S*. *pombe*, *A*. *nidulans*, *A*. *fumigatus* and *F*. *graminearum* [5,21,22,26,27]. However, Zhao et al. (2022) found that AtfA has no impact on the osmotic stress response in *A*. *flavus* strain NRRL 3357 [58]. Contrast to reported *AtfA* in *A*. *flavus*, we found that *AflatfA*, but not *AflatfB*, is involved in the response to osmotic stress in this study. Different recipient strain, osmotic reagents and selection marker RFP protein result in different effects of *AftA* on the osmotic stress response between this study and the previous study [58]. However, unlike the more sensitive to osmotic stress agents in above reported fungi, the *A*. *flavus* Δ*AflatfA* strain is more tolerant to NaCl, KCl and Sorbitol. It is noticeable that the osmotic stress response is not regulated by *Bcatf1* in *B*. *cinerea* [17]. Consequently, the osmotic stress sensing of Atf1/AtfA may demonstrate that AtfA/Atf1 may be not conserved in the osmotic stress response in fungi, and the regulatory mechanism is species-specific. The tolerance of *AflatfA* and *AflatfB* double mutant was increased to NaCl, KCl and Sorbitol, which is same to that of Δ*AflatfA* mutant, indicating that *AflatfA* and *AflatfB* may have no overlapping function in response to osmotic stress. Moreover, deletion of *AflatfA*, but not *AflatfB,* resulted in higher sensitivity to heat stress in *A*. *flavus*, which is similar to that in *S*. *pombe*, *C*. *neoformans*, *A*. *nidulans* and *A*. *fumigatus* [11,26,27,30,59]. Meanwhile, we found that the change trend of sensitivity to theΔ*AflatfA*Δ*AflatfB* strain to heat stress is the same as that of the Δ*AflatfA* mutant, suggesting that the response to heat stress is not regulated by *AflatfB*. Moreover, as the incubation time increases at 4 °C, the viability of conidia from the Δ*AflatfA* mutant dramatically decreased in *A*. *flavus* in comparison with WT, which is similar to *A*. *fumigatus* [27]. However, the phenomenon is not observed in the Δ*AflatfB* mutant. Interestingly, the germination rate of the Δ*AflatfA*Δ*AflatfB* mutant is higher than that of the Δ*AflatfA* mutant, but lower than that of WT. Thus, in fungi, the transcription factor *AtfA,* but not *AtfB,* might play a central role in temperature stress responses.

In fission yeast and filamentous fungi, *Atf1*/*AtfA* orthologs mediate vegetative growth and sexual and asexual development of fungi. Firstly, AflatfA contributes to the colony growth in *A*. *flavus*. Although Atf1/AtfA is conserved in the effects on radial growth, differences exist among fungi. In contrast to the moderate radial growth in *A*. *nidulans* [25,29], the lack of *Atf1*/*AtfA* led to a decreased colony diameter not only of *A*. *flavus* in this study on PDA and MM [58], but also of *S*. *pombe*, *F*. *verticillioides* and *M*. *oryzae* in previous studies [6,18,19]. Nevertheless, the lack of *AtfB* as an *AtfA* paralogous gene showed no obvious change of colony diameter on PDA compared to WT in this study; however, deletion of *AflatfB* brought about a decreased colony diameter in *A*. *flavus* NRRL3357 [58], suggesting that the effect of AtfB on vegetative growth is related with the medium type. Noticeably, we observed that the change trend in radial growth of the Δ*AflatfA*Δ*AflatfB* strain is the same as that of Δ*AflatfA* but not Δ*AflatfB*. Therefore, AflatfA and AflatfB both regulate colony growth, but the mechanism is complex. Moreover, similar to how the deletion of Atf1/AtfA homologous genes resulted in dramatically less conidia in fungi *A*. *nidulans*, *B*. *cinerea* and *V*. *dahliae* on an artificial medium [17,23,25,29], the Δ*AflatfA* mutant has a defect in conidia production not only on the artificial medium MM [58], but also on host plants such as corn and peanut in this study. A further study revealed that the germination ratio of the conidia of Δ*AflatfA* had declined. The deletion of the *AflatfB* gene showed no change to conidia production and the germination ratio of conidia, whereas the change trend of conidia production and its germination ratio of Δ*AflatfA*Δ*AflatfB* is similar to that of the Δ*AflatfA* mutant. The loss of *AflatfA* results in a strongly increased production of sclerotia in *A*. *flavus*. However, deletion of *AtfA* in *A*. *flavus* NRRL 3357 results in no sclerotia formation [58]. Conversely, the formation of sclerotia is both damaged in the Δ*Cpatf1* mutant of *C*. *purpurea* and in the Δ*Bcatf1* mutant of *B*. *cinerea* [17,24]. Therefore, Atf1/AtfA orthologs of fungi play an important role in regulation of sclerotia formation, but the specific regulatory mechanism is dependent on the species and strain. Similar to the reported *AtfB* in a previous study [58], we reported that AtfB positively regulates the sclerotia formation in *A*. *flavus*. In contrast to the impact of *AflatfA* on sclerotia formation, the Δ*AflatfB* mutant showed defects in sclerotia formation. Interestingly, as with single deletion of *AflatfA*, the double deletion of *AflatfA*/*AflatfB* brought about an increased production of sclerotia. Thus, it can be seen that AflatfA plays an opposite role to *AflatfB* in the regulation of sclerotia formation, and the function of *AflatfA* is decisive, which is similar to Atf1/AtfA orthologs in *S*. *pombe*, *A*. *nidulans*, *F*. *verticillioides* and *M*. *oryzae* [6,18,19,25,29]. Thus, it can be seen that AflatfA involves mycelium growth, conidia germination and sclerotia formation, whereas AflatfB only the regulates sclerotia formation.

## 4. Materials and Methods

### 4.1. Fungal Strains and Culturing Conditions

*A*. *flavus* strains listed in Table 1 were used or generated in this study. The Czapek–Dox Medium supplemented with 10 mM uridine and 10 mM uracil as needed was used to construct the mutants. For spore collection or RNA and DNA extraction, strains were cultured on PDA at 37 °C for 5 d or 29 °C for 48 h. To assess the colony and growth characteristics, both WT strain and mutants were maintained on YES or GMM with or without different chemical agents for 5 d at 29 °C in dark. As for the oxidative stress conditions, mycelium sensitivity to oxidative agents was measured by YES media supplemented with 10 mM H_2_O_2_. To observe the vegetative growth under osmotic stress conditions, the strains were inoculated on GMM containing 1.5 M NaCl, 1.5 M KCl or 1.4 M Sorbitol, and then the colony diameters were measured after 5 d. The sensitivity to temperature stress was analyzed on GMM after spores were treated by different temperatures. All strains as glycerol stocks were maintained at −80 °C. 

### 4.2. Identification, Gene Composition and Phylogentic Analysis of AflatfA and AflatfB

The protein sequence of AtfA (GeneID: 5991817, locus tag: AO0900003000685; Accession no.: XP_001824132) and AtfB (GeneID: 5996391, locus tag: AO090120000418, Accession no.: XP_001824132) from *A*. *oryzae* as the query were used to perform BLAST search to find the homologous genes in *A*. *flavus*. The AtfA and AtfB gene CDS region and upstream and downstream 2000 bp were loaded from the NCBI database. Then, the sequences were submitted to softberry web server, and the gene composition and protein sequence were re-predicted. We used the InterProScan 5 on EBI web server to predict the protein domain, including Atf1 from *S*. *pombe* (accession no. BAA09841.1), AoatfA from *A*. *oryzae* (accession no. XP_001819834.1), AfatfA from *A*. *fumigatus* (accession no. XP_754486.2), AtfA from *A*. *terreus* (accession no. EAU35111.1), AnAtfA from *A*. *nidulans* (accession no. ANN75015.1), PmAtfA from *T*. *marneffei* (accession no. KFX50645.1), ATF-1 from *N*. *crassa* (accession no. KFX50645.1), VdATF-1 from *V*. *dahliae* (accession no. XP_009621680.1), CPTF1 from *C*. *purpurea* (accession no. CCE33955.1), FgATF1 from *F*. *graminearum* (accession no. XP_011319081.1), MoATF1 from *M*. *oryzae* (accession no. ELQ67298.1) and Bcatf1 from *B*. *cinerea* (accession no. XP_024550682.1). The domains were visualized using IBS software. The protein sequences of AtfA homologous from fungi functionally verified by experiments were aligned with AtfA from *A*. *flavus* with Clustal X 2.0. A maximum likelihood tree was generated using a JTT Matrix model with 1000 replicates for bootstrapping by MEGA X software [61].

### 4.3. RNA Isolation and Reverse-Transcription PCR of AflatfA and AflatfB

*A*. *flavus* WT, Δ*AflatfA* and Δ*AflatfB* strains were inoculated on cellophane sheets laid on the PDA plates and cultured on media at 29 °C, and then the mycelia were harvested after 36 h. Then, total RNA was extracted using TRIzol Reagent, and DNA was digested with RNase-free DNase I. Subsequently, RNA was reverse transcribed into first-strand cDNA using the RevertAid First Strand cDNA Synthesis Kit (ThermoFisher Scientific, Waltham, MA, USA). To explore the transcription levels between the *AflatfA* and *AflatfB* genes, the sequence of the gene coding region was amplified using pairs 031340RT and 94010RT. The house-keeping gene actin was used as the internal control. The sequence of primers are listed in Appendix A.

### 4.4. Gene Deletion and Complementation of AflatfA and AflatfB

For constructing the *AflatfA* and *AflatfB* genes replacement, 1.1-kb upstream and 1.1-kb downstream fragments of a part of *AflatfA* and 1.1-kb upstream and 1.1-kb downstream fragments of a part of *AflatfB* were amplified from *A*. *flavus* genomic DNA, respectively. Meanwhile, the orotidine-5′-monophosphate decarboxylase (*pyrG*) gene was amplified from *A*. *fumigatus* gDNA. Then, the two flanking sequences of *AflatfA* and *pyrG* were joined together by overlap PCR with the primer pairs 31340 5F/F and 31340 3F/R, and the two flanking sequences of *AflAtfB* and *pyrG* were joined together by overlap PCR with the primer pairs 094010 5F/F and 094010 3F/R. Then, both approximately 3.8-kb segments were purified and transformed into the *A*. *flavus* recipient strain,. To construct the double deletion mutant, the gene knockout cassette comprising 2-kb of upstream and downstream fragments of *AflatfB* and a 1.8 kb of *ptrA* gene was generated by fusion PCR. Subsequently, the purified fragment was transformed into the *A*. *flavus* Δ*AflatfA* strain. To generate the complementation of the *AflatfA* and *AflatfB* mutants, the fragment of the ORF and its downstream and upstream flanking sequences of *AflAtfA* and *AflAtfB* were amplified and transformed into Δ*AflatfA* and Δ*AflatfB* strains, respectively. Then, the putative mutants were selected after being cultured on YGT containing 5-fluoroorotic acid (5-FOA) at 37 °C for 5 d. Subsequently, 1.0-kb fragment containing 500 bp terminator and its upstream 500 bp and 1.0-kb downstream fragments of the terminator were amplified from *A*. *flavus* genomic DNA. Then, two flanking sequences were fused with the *pyrG* gene amplified from *A*. *fumigatus* genomic DNA. At last, two purified segments were transformed into the Δ*ku70*Δ*pyrG*Δ*AflatfA*::*pyrG*, *AflatfA* strain and the Δ*ku70*Δ*pyrG*Δ*AflatfB*::*pyrG*, *AflatfB* strain to generate the complementation of the *AflatfA* and *AflatfB* deletion mutants. The preparation of *A*. *flavus* protoplasts and fungal transformation were both performed as the described method in the reference [62]. All mutants were firstly verified using diagnostic PCR with primers inside and outside the corresponding gene (Appendix A). Then, Southern blot was used to further verify the disruption strains.

### 4.5. Observation of Developmental Phenotype

To observe the radial colony growth, wild type and mutants were inoculated on PDA for 5 d at 29 °C. Conidia were harvested from 5-day-old colonies grown on PDA in 5 mL of distilled water containing 0.1% Tween 80, and then conidia were counted using a hemocytometer. Then, two hundred freshly grown conidia suspended in 2 μL 0.1% Tween 80 were inoculated. After 5 d, the colony diameters were measured, and the pictures were taken at the same time. To investigate the sclerotia production, 10^4^ fresh conidia (5 d) suspended in 2 μL 0.1% Tween 80 were spotted on modified Wickham medium plates. The plates were incubated in the dark at 37 °C for 12 d. To help the enumeration of sclerotia, 70% ethanol was used to kill and wash away conidia. Then, seven 7 mm-diameter agar plugs were taken from each plate, and the number of sclerotia was counted. Three biological replicates were included in each treatment, and whole experiment was repeated two times.

### 4.6. Oxidative Stress Bioassay

For the oxidative stress response assay of *A*. *flavus* conidia, 2 M H_2_O_2_ was added to conidia suspensions (H_2_O_2_ final concentration 100 mM and 200 mM respectively). The suspensions were immediately diluted to 10^5^ conidia/mL after 20 min of incubation at room temperature. In total, 200 spores suspended in 100 μL 0.01% Tween 80 were spread onto GMM agar plates, and were incubated at 29 °C for 48 h. Then, the number of colonies were counted, and the germinating rate was calculated. To test the H_2_O_2_-sensitivity of *A*. *flavus* mycelia, fresh conidia (5 d) were pipetted onto cellophane sheets (circle-shaped, 2-cm-diameter) laid on the YES agar plates and cultured for 24 h in the dark. Mycelial mates with sheets were transferred onto fresh YES or YES supplemented with 10 mM H_2_O_2_. Then, all plates were incubated at 29 °C, and the colony diameters were measured after being incubated for two to three days. All treatments included three replicates, and whole experiments were repeated two times.

### 4.7. Osmotic Stress Bioassay

To investigate the osmotic stress sensitivity of all mutants, the osmotic reagents NaCl, KCl and Sorbitol were chosen and added into the GMM medium. The concentration of osmotic reagents was as follows: 1.5 M NaCl, 1.5 M KCl or 1.4 M Sorbitol. Two hundred conidia were suspended in 2 μL 0.01% Tween 80 and spotted onto GMM plates with or without the abovementioned concentration reagents. All plates were incubated at 29 °C in dark for 5 d. From the second day to the fifth day, the colony diameters were measured. All treatments included three replicates, and whole experiments were repeated two times.

### 4.8. Temperature Treatment Assay

Conidia suspension of each strain was diluted to 10^5^ conidia/mL for the heat stress assay and the survival rate of the conidia upon incubation at the 4 °C assay. To analyze the heat stress response, 200 μL diluted conidia suspensions from each strain in 1.5 mL tube were incubated on block heaters at 50 °C for 1 h. Then, the tubes were immediately transferred and cooled on ice. The conidia were diluted to 2×10^3^ conidia/mL, and 100 μL conidia suspension was transferred and spread onto the GMM medium. After incubation at 29 °C in the dark for 48 h, the number of colonies was counted. Conidia without heat stress treatment were used as control. The germinating rate of conidia were calculated. To test the viability during preservation at 4 °C, the diluted suspensions (200 μL) from each strain were kept at 4 °C for 7 d and 14 d. As per the above description, the survival rates were calculated. All treatments included three replicates, and whole experiments were repeated two times.

### 4.9. Aflatoxin B1 Production Analysis

To measure the aflatoxin B1 production in vitro, 200 spores of WT, *AflatfA* and *AflatfB* mutant strains were spotted on a plate (*Φ* = 60 mm) containing 10 mL artificial YES medium (15% sucrose, 2% yeast extract, and 1% soytone), and cultured at 29 °C. After 5 d, six 7-mm-diameter agar plugs were taken from each plate, and transferred to an Eppendorf tube. Then, 3 mL dichloroethane was added into each plate, which were then sonicated for 1 h at room temperature. In total, 2 mL of extract was transferred to a new Eppendorf tube, and dried completely at room temperature. A total of 50 μL dichloroethane was used to suspend the dry extract, and 5 μL AFB1 standard (1 mg/mL) and extract were applied to TLC plates for analysis. A dichloroethane: acetone (90: 10, vol/vol) solvent system was used to develop TLC plates, and the bands were visualized under 254 nm light. 

### 4.10. Virulence Assays on Corn and Peanut

The infection experiments of corn and peanut were mainly performed as per the method invented by Chang et al. [63]. The following steps were modified. The eight seeds as a group were placed onto a Petri dish plate (*Φ* = 60 mm). Subsequently, a suspension of 3000 freshly harvested conidia in 3 μL of WT and all mutants were inoculated onto the surface of each seed. The seeds were spotted with 3 μL 0.01% Tween 80 solution as the controls. All seeds from each plate were transferred to tubes, and 10 mL of the 0.02% Triton-X 100 water was added into each tube. After being vortexed vigorously for 2 min to dislodge conidia, 500 μL suspension from each tube was used to count conidia using a hemocytometer, and the rest of the suspension was extracted with 8 mL dichloroethane. Subsequently, the extracts were shaken at 180 rpm/min for 30 min, and 5 mL extracts from every tube were transferred to a new tube. After being dried completely at room temperature, the extracts were resuspended in 100 μL dichloroethane. In total, 5 μL AFB1 standard (1 mg/mL) and extracts from each tube were applied to the TLC plates, respectively. The dichloroethane: acetone (90: 10, vol/vol) solvent system was used to develop the TLC plates, and TLC plates were visualized under 254 nm light.

### 4.11. Statistical Analysis

For statistical analysis, data were analyzed using the GraphPad Instat software package, version 5.01 (GraphPad software Inc.), according to the Tukey–Kramer multiple comparison test at *p* < 0.05. The different letters indicate the significant difference between data.

## Figures and Tables

**Figure 1 toxins-14-00857-f001:**
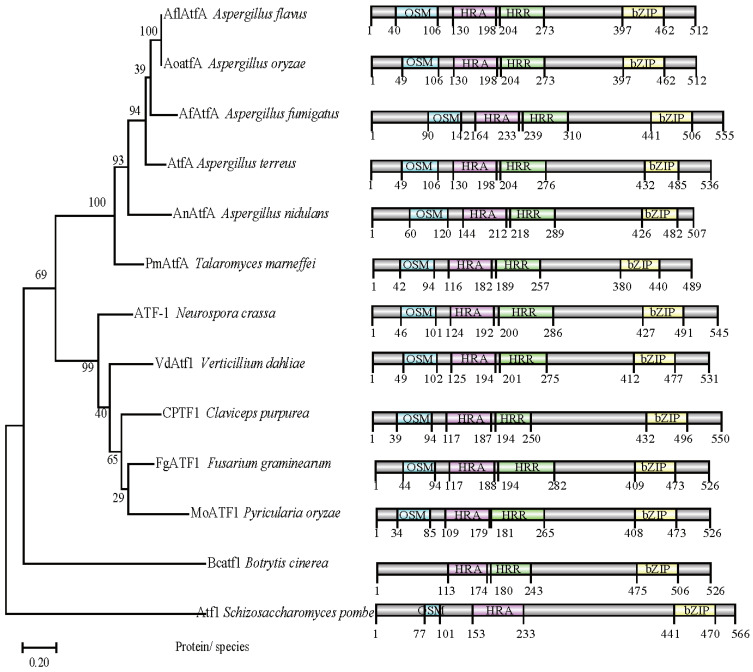
The phylogenetic analysis and functional domain of *Aspergillus flavus* transcription factor AflAtfA. Phylogenetic analysis of bZIP-type TF AtfA/ATF-1 from *A*. *flavus* and AtfA/ATF-1 orthologs that have been functionally verified in different fungi (Left). The protein sequences were aligned with Clustal X and the maximum likelihood tree was generated using MEGA X software. AtfA from *A*. *flavus* is in bold. *S*. *pombe* Atf1 (BAA09841.1), *A*. *oryzae* AoatfA (XP_001819834.1), *A*. *fumigatus* AfatfA (XP_754486.2), AtfA from *A*. *terreus* (EAU35111.1), *A*. *nidulans* AnAtfA (ANN75015.1), *T*. *marneffei* PmAtfA (KFX50645.1), *N*. *crassa* ATF-1 (KFX50645.1), *V*. *dahliae* VdATF-1 from (XP_009621680.1), *C*. *purpurea* CPTF1 (CCE33955.1), *F*. *graminearum* FgATF1 (XP_011319081.1), *P*. *oryzae* MoATF1 (ELQ67298.1) and *B*. *cinerea* Bcatf1 (XP_024550682.1) were used. The functional domain of *A*. *flavus* transcription factor *AflatfA* (Right). bZIP: basic leucine zipper domain; OSM: osmotic stress domain; HRA: domain that activates reorganization during meiosis; HRR: domain that inhibits reorganization during meiosis. The numbers at both ends of the protein indicate the length of each protein, whereas the numbers in the middle of the protein indicate the position of the domains. The protein domains are manually annotated with InterProScan on the EBI website.

**Figure 2 toxins-14-00857-f002:**
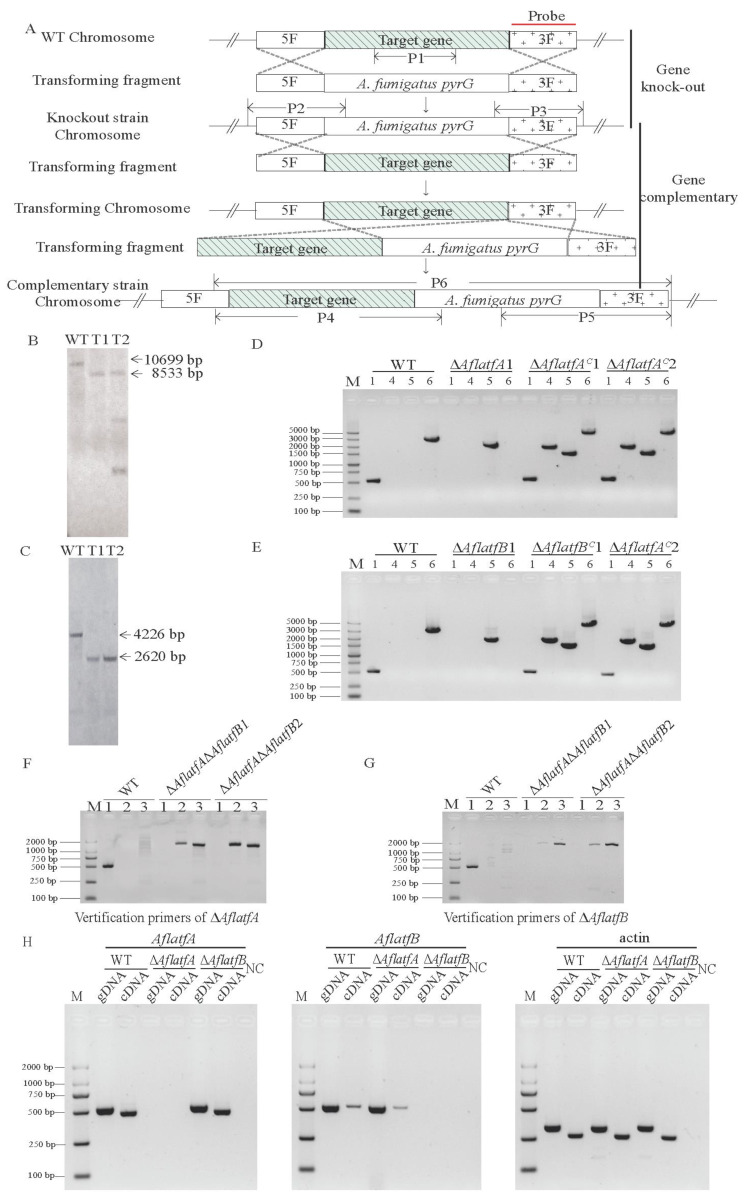
Construction of *AflatfA* and *AflatfB* mutant strains and analysis of the relationship between bZIP transcription factors *AflAtfA* and *AflAtfB*. (**A**) Construction strategy of *AflatfA* and *AflatfB* mutants and schematic diagram of PCR identification. The pair of primers P1_F/R was used to amplify the target gene. The pairs of primers P2_F/R and P3_F/R were used to amplify the knock-out mutants. The pairs of primers P4_F/R, P5_F/R and P6_F/R were used to amplify the complementary strains. (**B**) Southern blot verification of *AflatfA* deletion strains using 3′ flanking region as a probe. Genomic DNA from wild type and mutant strains were digested with *Kpn* I. The expected size is 10 699 bp and 8 533 bp for WT and mutant strains, respectively. (**C**) Southern blot verification of *AflatfB* deletion strains. The 3′ flanking region is used as a probe, and genomic DNA from wild type and mutant strains are digested with *Hind* III. The expected size is 4226 bp and 2620 bp for WT and mutant strains, respectively. (**D**) PCR was carried out to confirm the complementary strains of *AflatfA*. (**E**) PCR was carried out to confirm the complementary strains of *AflatfB*. (**F**,**G**) PCR verification of Δ*AflatfA*Δ*AflatfB* double knock strain using primers to verify Δ*AflatfA* and Δ*AflatfB* strains (**H**) Transcription levels of *AflatfA* and *AflatfB* genes and internal reference gene actin were analyzed in WT, Δ*AflatfA* and Δ*AflatfB* genomes by RT-PCR. gDNA, genomic DNA; cDNA, reverse transcribed DNA; NC, negative control; the letter M indicates DNA marker. The number from Figure 2D–G indicates the primer pair marked in Figure 2A.

**Figure 3 toxins-14-00857-f003:**
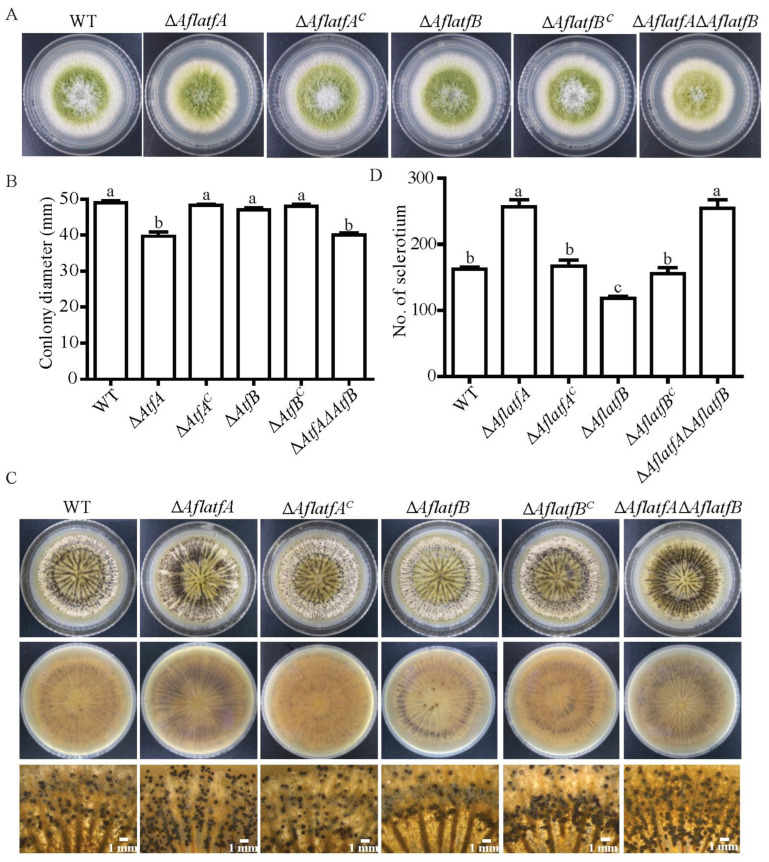
The function of AflatfA and AflatfB on mycelial growth and the production of sclerotia in *Aspergillus flavus*. (**A**) Mycelial growth of wild type, AflatfA and AflatfB mutant strains. Two hundred conidia of WT and mutants were cultured on PDA medium at 29 °C for 5 d. (**B**) Comparison of colony diameter between wild type and mutant strains on the fifth day. (**C**) Visual phenotype of sclerotia produced by wild type, *AflatfA* and *AflatfB* mutants. 10^3^ conidia/plate was incubated on Wickerham medium and cultured at 37 °C for 12 d. (**D**) Statistical analysis of sclerotia yield from wild type and mutant strains. Two independent biological experiments were performed with three replicates each time. Error bars represent the standard deviations. The different lowercase letter means that the difference between treatments is significant at *p* < 0.05.

**Figure 4 toxins-14-00857-f004:**
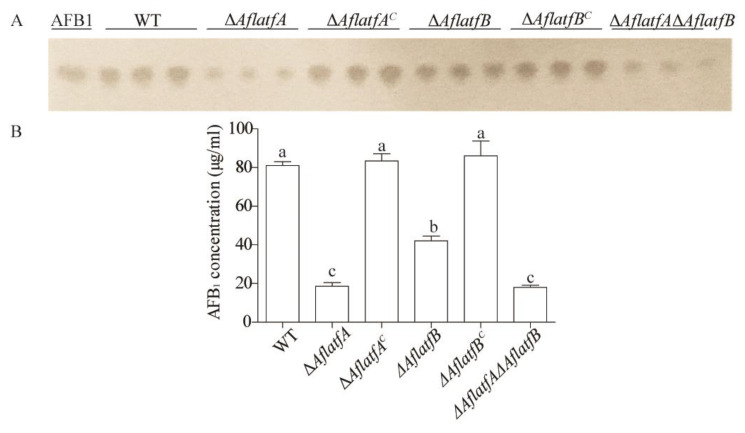
The bZIP transcription factors AflatfA and AflatfB in *Aspergillus flavus* positively regulate the production of aflatoxin B1. (**A**) Thin layer chromatography analysis of aflatoxin B1 produced by wild type, *AflatfA* and *AflatfB* mutant strains on artificial YES medium. AFB1 = aflatoxin B1 standard. (**B**) Statistical analysis of aflatoxins B1 produced by wild type, *AflatfA* and *AflatfB* mutants on YES artificial medium. Two independent biological experiments were performed with three replicates each time. Error bars represent the standard deviations. The different lowercase letter means that the difference is significant at *p* < 0.05.

**Figure 5 toxins-14-00857-f005:**
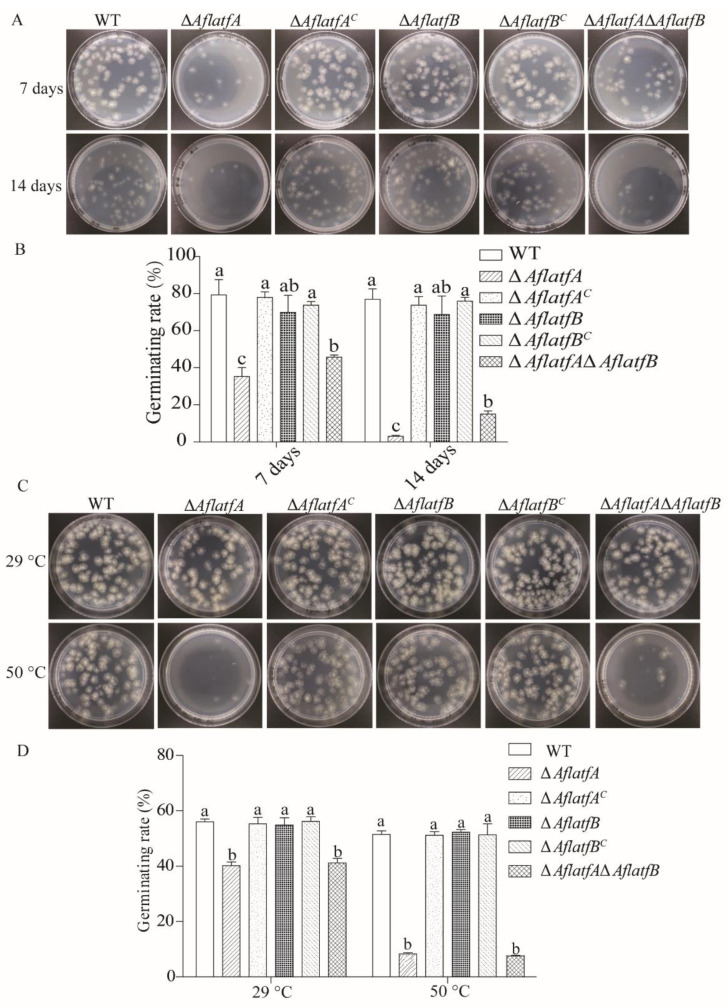
The effect of AflatfA and AflatfB genes on the survival rate of *Aspergillus flavus* stored at 4 °C and treated by high temperature. (**A**) Germination of wild type, *AflatfA* and *AflatfB* mutants after storage at 4 °C for 7 d or 14 d. Two hundred conidia were spread on GMM medium and cultured for 48 h. (**B**) Statistical analysis of survival rates of conidia of wild type and mutant strains after storage at 4 °C for 7 d or 14 d. (**C**) The stress tolerance against heat stress was examined. Conidia suspensions were heated at 55 °C or incubated at 29 °C as control for 1 h. Two hundred conidia were plated onto GMM plates, and visible colonies were counted after incubation at 29 °C for 48 h. (**D**) Statistical analysis of germination rates of conidia after heat treatment. Two independent biological experiments were performed with three replicates each time. Error bars represent the standard deviations. The different lowercase letter means that the difference is significant at *p* < 0.05.

**Figure 6 toxins-14-00857-f006:**
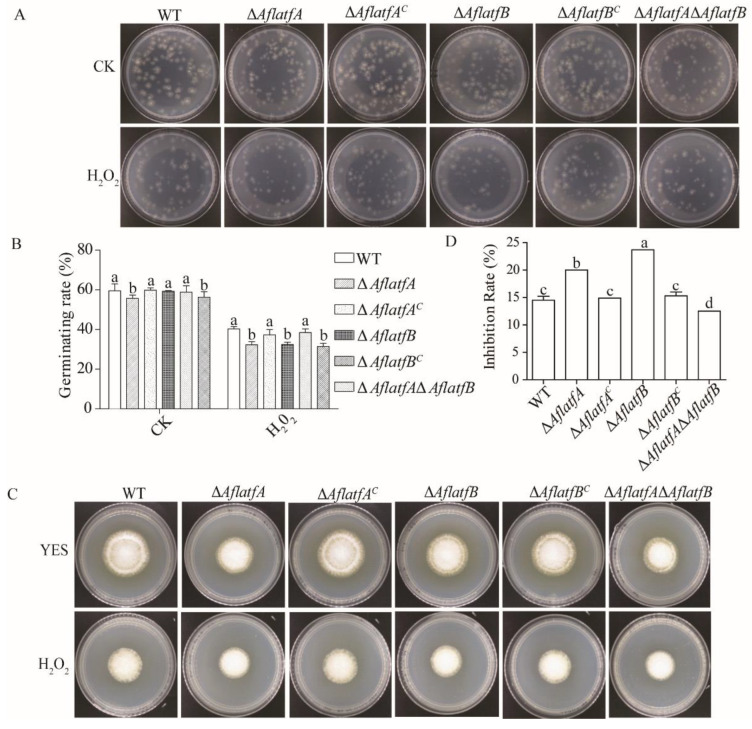
Hypersensitivity of *AflatfA* and *AflatfB* mutants to oxidative stress. (**A**) The stress tolerance of conidia against oxidative stress was examined. Conidia suspensions were incubated at room temperature with 200 mM H_2_O_2_ for 20 min. Two hundred of conidia were plated onto GMM plates, and visible colonies were counted after incubation at 29 °C for 48 h. (**B**) Statistical analysis of germinating rate of conidia from WT and mutants on YES under oxidative stress. (**C**) The stress tolerance of mycelium against oxidative stress was examined. Equal conidia were pipetted onto cellophane sheets on the YES plates and cultured at 29 °C for 24 h. Mycelial mates with sheets were transferred onto fresh YES plate with or without 10 mM H_2_O_2_ and cultured at 29 °C for 3 d. (**D**) Statistical analysis of mycelium growth of WT and mutants on YES under oxidative stress. Measurements of growth inhibition rate are relative to growth rate of each untreated control. Two independent biological experiments were performed with three replicates each time. Error bars represent the standard deviations. The different lowercase letter means that the difference is significant at *p* < 0.05.

**Figure 7 toxins-14-00857-f007:**
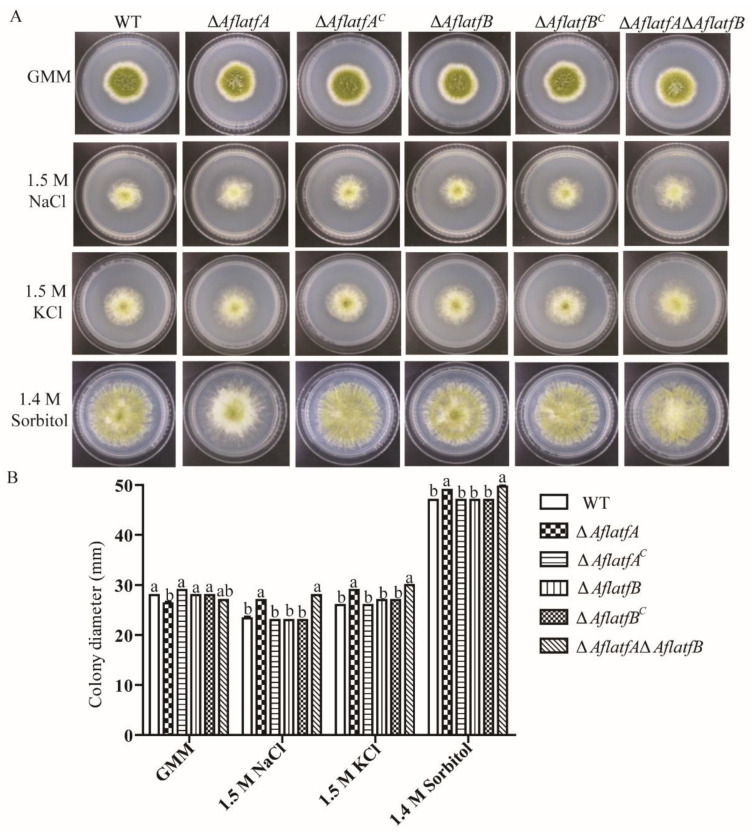
AflatfA response to osmotic stress. (**A**) Mycelia growth of the *A*. *flavus* wild type and *AflatfA* and *AflatfB* mutants under osmotic stress. Two hundred conidia of *A*. *flavus* WT and mutants were cultured on GMM media supplemented with or without NaCl (1.5 M), KCl (1.5 M) or Sorbitol (1.4 M) at 29 °C for 5 d. (**B**) Statistical analysis of colony diameter of the testing strains measured on the 5th day. Each treatment included three replicates. Error bars represent the standard deviations. The different lowercase letter means that the difference between treatments is significant at *p* < 0.05.

**Figure 8 toxins-14-00857-f008:**
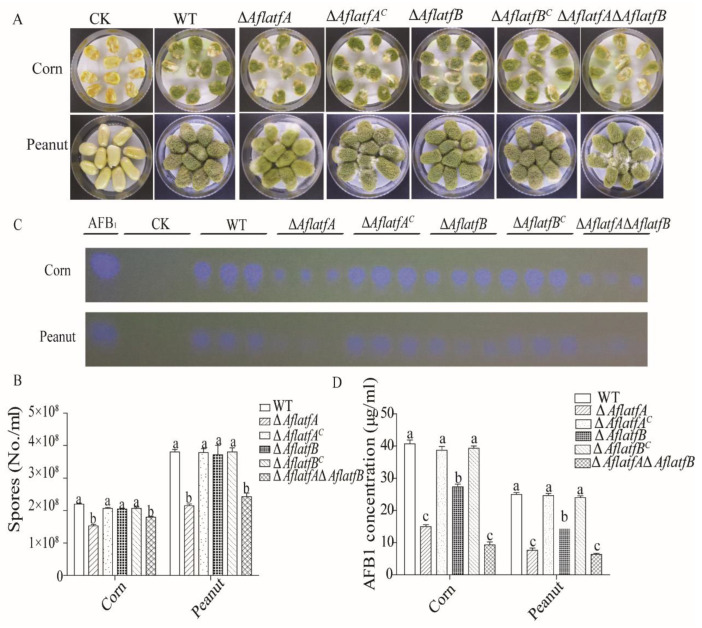
The pathogenicity of *Aspergillus flavus* on peanut and corn is regulated by AflatfA and AflatfB. (**A**) Growth of wild type, *AflatfA* and *AflatfB* mutants on corn and peanut after inoculation for 5 d. (**B**) Statistical analysis of conidia produced by wild type, *AflatfA* and *AflatfB* mutants on corn and peanut. (**C**) Thin layer chromatography analysis of aflatoxins extracted from corn and peanut. AFB1 = aflatoxin B1 standard. (**D**) Statistical analysis of aflatoxin B1 from corn and peanut infected by wild type, *AflatfA* and *AflatfB* mutant strains. Two independent biological experiments were performed with three replicates each time. Error bars represent the standard deviations. The different lowercase letter means that the difference is significant at *p* < 0.05.

**Table 1 toxins-14-00857-t001:** Fungal strains used in this study.

Strain	Description	Reference
Recipient strain	PTSΔ*ku70*Δ*pyrG*	[60]
wild type (WT)	*PTS*Δ*ku70*Δ*pyrG::AfpyrG*	[60]
Δ*AflatfA*	Δ*ku70*Δ*pyrG*Δ*AflatfA::pyrG*	This study
Δ*AflatfB*	Δ*ku70*Δ*pyrG*Δ*AflatfB::pyrG*	This study
Δ*AflatfA^C^*	Δ*ku70*Δ*pyrG*Δ*AflatfA::pyrG*, *AflatfA::pyrG*	This study
Δ*AflatfB^C^*	Δ*ku70*Δ*pyrG*Δ*AflatfB::pyrG*, *AflatfB::pyrG*	This study
Δ*AflatfA*Δ*AflatfB*	Δ*ku70*Δ*pyrG*Δ*AflatfA::pyrG*Δ*AflatfB::ptrA*	This study

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
