# Peer review of "Genetic Interaction of Global Regulators AflatfA and AflatfB Mediating Development, Stress Response and Aflatoxins B1 Production in Aspergillus flavus"

_toxins, 2022, doi:10.3390/toxins14120857_

Round 1
Reviewer 1 Report
The manuscript entitled “Genetic interaction of global regulators AflatfA and AflatfB mediating development, stress response and aflatoxins B1 production in Aspergillus flavus” is very interesting. I believe this study has an applicable significance in the field of food storage research and development. The study was well written and designed.
1. The figure 2 D-G, the number under each strain are representing the repeats or different conditions? Please clarify and mention them in the figure legends.
2. I recommend marking the ladders to indicate the band size (bp).
3. The densitometric analysis is required for figure 2H to conclude that transcriptional level of AflatfB was lower in the genome of AflatfA deletion strain, compared to WT strain; whereas, deletion of AflatfB has no impact on AflatfA transcriptional level. Because the results do not seem to be significantly different.
4. Please indicate the scalebar in Figure 3C (microscopic image).
5. Results demonstrated in figure 5 are very interesting. Authors observed that long term storage of both ΔAflatfA and ΔAflatfAΔAflatfB strains at 4 degree significantly decreased its germination potential. Keeping these results in mind, it would be interesting to see if the germination rate of these mutants’ strain on peanuts and corn at 4 degree for 14 days differs or not, compared to WT strain.
6. Provide the sequence of primers used in the study.
7. I strongly encourage to improve the introduction and discussion section of the manuscript. Write how these mycotoxins are incorporated in the food chain and being toxic to the living organisms present in the environment. The below mentioned papers are suitable for citation:
Wen et al., 2005. Appl Environ Microbiol. PMID: 15933021.
Det et al., 2022. CritRev Food Sci Nutr. PMID: 35445609.
Reviewer 2 Report
Authors of this paper performed a classical gene study on aftA and aftB in Aspergillus flavus. With the exception of clarifications on how certain experiments were performed I believe the results of the manuscript to be valid. That said I do not think there is enough novelty in the study as it is. The previous transcription factor published by Zhao (et al, 2022) covers similar ground by creating deletion mutants of atfA and atfB and testing various phentoypes such as osmotic stress, secondary metabolism, and virulence. Authors have expanded on research of these genes by creating a double mutant and performing additional experiments. While additionally some of the results disagree with the previous study, I think more is required of this study. I would strongly recommend and encourage author to look at how atfA and atfB influence the expression of genes involved in sclerotia production and oxidative stress. I do like the inclusion of the double mutant to look at the relationship between atfA and aftB and should in include in gene expression studies.
Other comments:
· Line 18 change “less sensitive” to “less sensitivity”
· Line 21 Sentence “When on corn and peanut….” is confusing and not sure what is producing less conidia comparted to the WT
· Line 71 Fungicide effect? Do you mean resistance?
· As it seems these genes have already been studied in A. flavus (Zhao et al., 2022) it would be appropriate to follow the same naming convention for the genes and refer to them as aftA and aftB as to avoid the notion of these being different genes.
· General mention of the figures I don’t like the labeling of figures where the letters are out of order. For example, in figure 2, Panel F & G appear before D & E. My recommendation is to split these figures for better sizing or reorganize them.
· Please include AtfA and AtfB IDs from A.oryzae in paper that were used to identify homologues in A. flavus.
· Where is the phylogenetic analysis for atfB?
· I like Figure 2A but with the complexity of the rest of the figure the strategies for knocking out both atfA and atfB needed. For example, showing where the primer sites for the southern probes are in relationship to the Kpn1 and HindIII sites are would be helpful. I think an additional supplemental figure is needed to explain how the complementation strain was made. Only after reading the relevant section in the materials and methods multiple times did I start to understand what was done.
· I think more explanation of what the different numbers are indicating in panels D, E, F, & G of Figure 2 is needed.
· Please demonstrate the gene expression for atfA and atfB in the double mutant akin to what was done in 2H and 2M.
· Figure 2, label for M is repeated twice for atfB and actin. Also this should be changed to I and J respectively.
· Line 224 says the following “…AflatfA plays a more important role function than AflatfB for sclerotia formation”. Why is this the case? I think the case could be made that atfA is upstream of atfB but it does not mean that it is more important with what is shown here.
· Based on quantity of sclerotia counted in figure 3, how did a 7mm plug generate so many sclerotia? Was this number used to extrapolate the amount of sclerotia on the entire plate? If so please specify.
· Figure 4. Panel A is very hard to view. Please adjust. For panel B no description is made of how Aflatoxin concentrations are determined add this information to the figure legend and materials and methods as appropriate. The same also holds true for the virulence experiments in Figure 8.
· In Materials and Methods section 4.3, a reference should be made to the primer table located in the supplemental files. This section is also missing information on how the qPCR itself was done.
Reviewer 3 Report
toxins- 2034645-peer-review-v1
Reviewer summary.
The goal of this work was to investigate and characterize the genetic interaction between stress response transcriptional regulators AtfA and AtfB in mediating fungal development, stress response, aflatoxin production, and virulence of Aspergillus flavus. The authors used a comprehensive set of approaches (genetic knockout and complementation, biochemical characterization, stress tolerance assays, and infection assays) to characterize AtfA and AtfB function in A. flavus.
Major findings demonstrate several novel and different functions from other Aspergillus spp. (A. parasiticus, A. oryzae) that were well characterized by this group
(1) AtfA negatively regulates sclerotia formation, AtfB positive regulates sclerotia formation (with AtfA having a stronger regulatory role)
(2) AtfA and AtfB positively regulate AFB1 levels, with AtfA demonstrating a greater magnitude of regulation than AtfB, there seems to be no synergistic effect of AFB1 regulation in the double mutant which is interesting
(3) relative to conidia response to temperature stress, AtfA is a dominant regulator of conidial viability and heat stress tolerance
(4) AtfA and AtfB regulates mycelium response to oxidative stress (possibly with AtfB having a larger regulatory role, Figure 6D; although no synergistic effect on conidia response to oxidative stress observed in the double mutant)
(5) Relative to osmotic stress, similar to temperature stress, AtfA is a dominant regulator of osmotic stress response (could be due to its role as a global regulator, binding to response elements that are more widely distributed in promoters of stress and temperature response genes
(6) Finally, the authors showed that AtfA but not AtfB significantly decrease spore load on corn and peanut and both single knockout strains and double mutant decrease AFB1 levels on corn and peanut.
Outcomes from this work highlighted the differential role of AtfA and AtfB in regulation of secondary metabolism, global stress response pathways, and virulence in Aspergillus flavus. This work continues to show while much is known about aflatoxin biosynthesis, variations in gene regulatory networks is dependent on species and strain and could be important to control aflatoxin contamination in the field and during storage.
I have highlighted some strengths in this study as well as some comments and suggestions for authors consideration.
Strengths.
- Authors demonstrated clear goals and motivation to conduct study
- Strength in approach, for example authors systematically and clearly design and conducted single mutant knockouts and double mutants, as well as complementation strains (the strength in this study is in characterization of 6 strains)
- Authors comprehensively characterized seven different endpoints for all strains used in this study
- Figures are outstanding, and figure legends are well written and concise
Title
Based on the results, it appears that AtfA but not AtfB is a global regulator – however authors call AtfA and AtfB global regulators in the title, may consider revising title
Abstract
L22-24 “Taken together, this study reveals that AflatfA controls more cellular processes and plays more important roles (this sentence is redundant) than AflatfB expect response to H2O2, which might result from effect of AflatfA on transcriptional level of AflatfB.” – sentence is not clear, consider rephrasing
Introduction
L46 activates
L52 cryptic
L57 atf1 should be italicize?
L71 fungicide effect – what does this mean, it would help reader if authors follow up with a sentence on what this means
L133 – plays a decisive role with no overlapping function (unsure what the authors mean here)
Results.
L144 cDNA
L251 unsure what YES artificial medium is
L256 – remove delta symbol
L264 consider rewriting sentence – Thus, AtfA but not AtfB mediates…
L274 consider rewriting sentence- ….implied that AtfA but not AtfB
L290 – how was 200 mM concentration of hydrogen peroxide determined?
L330 – The above results show that AtfA but not AtfB regulates osmotic stress response in A. flavus
L341 after infecting corn and peanut..
L349 when comparing…
Discussion.
L391 “fungal secondary metabolism is conserved but the regulatory effect of other pathways? is divergent”; regulatory effect – unsure what this means
L404 is it more important or more dominant (greater effect)
L462 ..observed trend changes in radial growth…
How can the outcomes of this study help the scientist control aflatoxin production? Future studies and current gaps –
Materials and Methods.
L493 – authors should state what it refers to
L497 sensitivity
L520 how was mycelium separated from agar for RNA isolation, grown on cellophane?
L523 To explore the..
L569 how were these concentrations determined, dose response?
Reviewer 4 Report
This manuscript elucidates the interaction between two key transcription factors in Aspergillus flavus. The authors look at the effect of single and double deletions on a wide variety of properties of the fungus, including mycelial growth, sclerotia formation, conidial viability, aflatoxin B1 production and reaction to a variety of osmotic reagents and osmotic stress.
The introduction is very complete and summarizes well what is already in the literature. It appears to be a follow-up study from a recent one (reference 56, 2022) but here looking at the interaction between the two transcription factors. The manuscript is well organized, with few, if any, errors/typos and treats each test of the single and double knockouts in order. There are eight figures, some of which are quite complex, but are required to illustrate the effect of each test. All tests of the effect of knockouts against the wild type are accompanied by correct statistics – two biological replicates with three replicates within and standard deviations are reported.
The only suggestion would be to reduce the discussion a bit and focus on the conclusions provided by this study as they get a bit lost in the comparisons to the literature (see below).
Specific comments:
Page 12, line 327: After “WT strain” – Isn’t this Figure 7 not 6?
Discussion: The discussion is very long and appears to summarize much of what is in the literature which was well explained in the introduction. While the authors treat each test in order and compare their new results to the literature, it gets a bit confusing to the reader as to what THIS particular study found. The authors could perhaps reduce this section somewhat by focusing on the results presented – e.g. page 14, lines 396-405, while lines 405-425 are good. It needs a concluding sentence as to what this study contributed to the role of these transcription factors AflatfA and AflatfB in Aspergillus flavus, similar to what is contained in the abstract which was well written.
Round 2
Reviewer 1 Report
The present format of the manuscript has been modified based on the comments given, which is satisfying. Therefore, I believe the manuscript has the potential to be published in its current format.
Author Response
Thank you!
Reviewer 2 Report
Authors have made great strides in the presentation and clarity of their work. The figure presented are much clearer and now streamline the communication of their work. All of my comments minus one have been addressed by the authors sufficiently.
One thing that still needs to be addressed is confirming the expression, or lack of expression, for AflatfB in the double mutant. Authors mention expression was not tested since were both genes were deleted. While diagnostic PCR was done to show successful transformants, no expression assays were performed to confirm that this strain is a true double deletion as done simliarly with deletion AflaftA and AflaftB. Lack of expression of AflatfA can be inferred for this strain due to its progenitor strain T1 being a AflatfA deletion that had its expression confirmed in figure 2. This was not done for AflatfB after creation of the strain. As many experiments and conclusions in this publication rely on this strain I think it would only strengthen the results to validate this aspect.
Reviewer 3 Report
The authors have addressed all of my comments and questions. Thank you!
Author Response
Thank you!